# Bacterial Microbiota in Soil Amended with Deoxynivalenol-Contaminated Wheat

**DOI:** 10.3390/toxins17120565

**Published:** 2025-11-22

**Authors:** Emmanuel W. Bumunang, Kim Stanford, Yuxi Wang, Benjamin Ellert, Matthew Waldner, Trevor W. Alexander

**Affiliations:** 1Department of Biological Sciences, University of Lethbridge, Lethbridge, AB T1K 1M4, Canada; emmanuel.bumunang@uleth.ca (E.W.B.); kim.stanford@uleth.ca (K.S.); 2Agriculture and Agri-Food Canada, Lethbridge Research and Development Center, Lethbridge, AB T1J 4B1, Canada; yuxi.wang@agr.gc.ca (Y.W.); benjamin.ellert@agr.gc.ca (B.E.); matthew.waldner@agr.gc.ca (M.W.)

**Keywords:** deoxynivalenol (DON), soil, microbial diversity, bacterial consortium, DON degradation

## Abstract

Feed contaminated with the mycotoxin deoxynivalenol (DON) can negatively impact livestock health and performance. Bacteria capable of degrading DON present a method of mitigating its harmful effects. This study aimed to identify microbial consortia from soil samples that could degrade DON. Soil from central (Lacombe, LA) and southern (Lethbridge, LE) Alberta were used as microbial inoculant. The soils were mixed with DON-contaminated wheat (18 ppm/kg) on day 0, and each soil type was divided into triplicate pots (180 g) and placed in a controlled environment for 32 d. Control pots of each soil type were included, which contained no DON-contaminated wheat. On days 0, 7, 14, and 32, 1 g subsamples were collected from pots, serially diluted in a limited medium containing DON (10 µg/mL) as the only carbon source, and incubated for 2 weeks (30 °C). DNA was extracted from the pots across time, as well as the subsample consortia grown in DON-amended medium, and was analyzed for bacterial changes after 16S rRNA gene sequencing. The relative abundance of bacterial genera in soil samples after enrichment with DON-contaminated wheat increased across time compared to the baseline day 0 time point. DON-degrading activity (26%) was only detected in LA soil suspension on day 7, and was highest after 14 days of incubation. The most abundant bacteria in the LA DON-degrading consortia belonged to the *Pseudomonas* (8.8%), *Delftia* (7.4%), *Acinetobacter* (6.4%), *Comamonas* (5.7%), *Stenotrophomonas* (5.5%), *Shinella* (5.5%), *Ensifer* (5.1%), *Agrobacterium* (5.0%), *Achromobacter* (4.7%), and *Rhizobium* (3.7%) genera. *Pseudomonas aeruginosa* (*n* = 9) and *Serratia liquefaciens* (*n* = 3) strains isolated from the LA consortia did not degrade DON. Overall, this study shows that the soil contained bacteria capable of degrading DON; however, variation existed depending on the soil’s source.

## 1. Introduction

Mycotoxins, such as deoxynivalenol (DON) produced by *Fusarium graminearum*, are harmful secondary metabolic products that cause economic losses and health issues to humans and livestock. According to the Manitoba Crop Alliance, *Fusarium* head blight (FHB) causes economic losses in three ways: (1) yield losses; (2) downgrading of grains; and (3) mycotoxin contaminations reducing grain marketing options [1]. *F*. *graminearum* is the most prevalent species causing FHB in Alberta [2], posing a potential threat to animal production. Therefore, measures to control and prevent feed contamination with DON are necessary to ensure the safety of livestock. These include strategies based on preventing *Fusarium* contamination of feed or limiting its metabolic products in feed through mycotoxin adsorption and degradation.

Soil is a valuable source of diverse microbial communities, potentially harboring DON-degrading bacteria that can be developed into feed additives to mitigate DON. For example, a microbial consortium from soil samples was able to transform DON into de-epoxy DON (DOM-1) [3]. Other bacterial consortia from soil have also been reported to transform DON to DOM-1 or 3-keto-DON [4,5]. DON degradation products are believed to be less toxic than the main toxin, and their transformations are dependent on strain(s), cultivation conditions (Aerobic vs. Anaerobic), and the sample source [6].

Microbial communities from various environmental sources may have a distinct response to DON enrichment, with some strains increasing or decreasing after DON supplementation [5,7]. The shifts in microbiota indicate that bacterial strains can utilize DON as a carbon source or biotransform it through cooperative catabolism [8,9]. The differential microbial response to DON may enable the development of effective DON biotransformation agents and applications, such as bacterial additives to mitigate contaminated feed. The present study aimed to identify microbial consortia in diverse soil samples from central (Lacombe, LA) and southern (Lethbridge, LE) Alberta that could degrade DON. These soils were enriched with DON-contaminated wheat and studied across time. The LE soil samples were from manure-amended (LE-MA) or non-manure-amended (LE-NM) plots to increase bacterial diversity of the soil investigated [10].

## 2. Results

### 2.1. Chemical Analysis of Soil Samples

The LA soil was slightly acidic (pH 6.4) compared to LE-MA (7.3) and LE-NM (pH 7.0), while moisture content was greater in LA (18.4%) than LE-MA (14.7%) and LE- non-MA (13.5%) soils (Supplemental Appendix A). Total carbon (38.6 mg g^−1^) and nitrogen (3.3 mg g^−1^) was greater in LA soil compared to LE-NM (21.5 and 1.8 mg g^−1^) and LE-MA (29.3 and 2.5 mg g^−1^). Lacombe soil is orthic black chernozem according to the Canadian system of soil classification, whereas Lethbridge soil is orthic dark brown chernozem.

### 2.2. Community Structure and Composition of Soil Bacteria

The microbial communities of the soil samples were analyzed by sequencing the 16S rDNA genes to determine the abundances of bacterial genera across different time points. Differences in alpha diversity metrics were observed. On day 0, for both the Shannon diversity index and richness, LE-MA was greater than LA and LE-NM (*p* < 0.05), while there were no differences between LA and LE-NM (*p* > 0.05) (Figure 1). A PERMDISP test revealed that microbial variability did not differ significantly by soil source (*p* = 0.4284). However, PERMANOVA indicated that bacterial structure of the microbiota differed significantly by the source of soil (*p* = 0.0062; R^2^ = 0.64367). Detrended correspondence analysis (DCA) plots supported the PERMANOVA results, indicating clustering by soil type, with LE-MA and LE-NM bacteria being more closely related compared to LA microbiota (Figure 2).

While variation existed, on day 0, the most relatively abundant genera were similar for LE-MA and LE-NM (Figure 3). The predominant genera in LE-MA soil were *Vicinamibacter* (6.20%), *Flavisolibacter* (4.18%), *Microvirga* (4.03%), *Planococcus* (2.52%), *Flavitalea* (2.15%), *Haliangium* (2.09%), and *Massilia* (1.86%), while the most abundant genera in LE-NM soil were *Vicinamibacter* (6.80%), *Microvirga* (5.97%), *Flavisolibacter* (3.27%), *Massilia* (2.51%), *Gaiella* (2.16%), and *Skermanella* (1.81%). In LA soil, the dominant taxa were *Alloacidobacterium* (6.38%), *Gaiella* (4.67%), *Gemmatimonas* (3.80%), *Bradyrhizobium* (3.71%), *Occallatibacter* (3.20%), and *Bryobacter* (2.49%).

### 2.3. Changes in the Microbiota Across Time

Across time, microbiota in the control LA soil pots did not change in abundance (*p* > 0.0001). However, for pots amended with DON-contaminated wheat, a total of 67 genera either increased or decreased in relative abundance compared to the day 0 baseline (Figure 4, *p* < 0.0001). Of these, the relative abundance of 31 genera increased while 36 decreased during at least one time point. Of the top ten genera present on day 0 (Figure 3), only *Massila* increased after supplementation of DON-contaminated wheat. For LE-MA control soil pots, 16 genera changed during at least one time point, with 13 genera increasing and three decreasing, compared to day 0 (Figure 5, *p* < 0.0001). In contrast, the LE-MA pots amended with DON-contaminated wheat had 34 genera changing in abundance, of which 25 increased and 9 decreased in abundance. All of the changes in abundance occurred on days 14 and 32. Similarly to LA soil, none of the taxa changed in LE-NM control pots across time, except for *Planomicrobium*, which decreased on day 32 (Figure 6, *p* < 0.0001). In contrast, the LE-NM pots amended with wheat had 37 genera increase or decrease in abundance between days 7–32, with most taxa increasing (20 genera) in abundance compared to day 0.

### 2.4. Changes in the Microbiota Within the LA Consortium

DON-degrading activity was only detected from LA soil samples (consortium LA), in M9 broth containing 10 µg/mL of DON as the sole carbon source, and was highest after two serial passages in which 26.4% of DON was degraded. No visible growth was observed in M9 broth containing 10 µg/mL of DON as a sole carbon source after three serial passages of incubation. The LA consortium did grow in BTSB containing 10 µg/mL DON with or without dextrose, but did not degrade DON.

To evaluate bacterial selection in the LA consortium, after growth with DON as a carbon source, the microbiota were compared to LA day 7 pots. In total, 19 genera changed in abundance when the consortium was compared to day 7 soil (Figure 7, *p* < 0.0001). Of these changes, ten genera increased in abundance in the consortium, including *Pseudomonas*, *Delftia*, *Acinetobacter*, *Comamonas*, *Stenotrophomonas*, *Shinella*, *Ensifer*, *Agrobacterium, Achromobacter*, and *Rhizobium.* It is noteworthy that *Delftia*, *Comamonas*, *Ensifer*, and *Shinella* were not detected in day 7 of the LA soil. To evaluate whether bacteria from the consortium that degrade DON could be isolated, the LA consortium grown in M9 (10 µg/mL of DON) was plated onto M9 agar for culturing bacteria. Randomly selected isolates were identified by 16S rRNA sequencing and belonged to *Pseudomonas aeruginosa* (*n* = 9) and *Serratia liquefaciens* (*n* = 3). None of the isolates degraded DON when cultured in M9 or BTSB (10 µg/mL of DON).

## 3. Discussion

Degradation of DON by microorganisms is increasingly being investigated for the mitigation of this mycotoxin, which poses a significant risk to livestock. Soil is a potential reservoir for diverse microbial communities, which can be exploited to mitigate DON. In this study, soil samples were collected from different locations, two of which grew barley infected with *F*. *graminearum*. Different soil types were utilized to increase the diversity of microbiota [11] and potential DON-degrading bacteria. Soil pH, moisture, and available nutrients alter the biodiversity of soil microbiotas and their respective biological activities [12]. Slightly acidic to neutral pH soils may enhance microbial activity compared to acidic or alkaline ones. As expected, the moisture, carbon, and nitrogen content of the soils varied in this study.

A bacterial community within a particular area is characterized by its species richness, population composition, and microbial diversity [13]. When evaluating bacteria, the LE-MA soil had the greatest diversity on day 0. While varying nutrient composition of the soils may have led to this difference, it is also possible that the manure used for fertilization contributed bacteria to the soil microbiota. Manure has previously been shown to alter the microbiota of soil for months after fertilization, both increasing diversity by altering nutrient composition and transferring bacteria from livestock to the soil, compared to non-manured soil [10]. Overall, the bacterial structure was more similar between the two Lethbridge soil types, compared to Lacombe, indicating that soil type impacts microbiota to a greater extent than manure fertilization.

The changes observed in the microbiota of potted soil across time indicate that DON-containing wheat amendment influenced the bacterial communities. This was apparent, as control soil pots not supplemented with wheat had relatively static microbiota, apart from the LE-MA soil. While a limitation of our study was not including treatment with non-contaminated wheat, it was likely that the noted changes were influenced mainly by metabolism of the wheat, and not DON. In support of this theory, most taxa that increased across time were common for all soil types, including *Acidovorax*, *Leclercia*, and *Pantoea*, which are commonly associated with soil and promote plant health [14,15]. In addition, DON metabolism was not detected in the Lethbridge soils, suggesting that bacterial changes were not from bacterial utilization of DON in those soils. Interestingly, Enterobacter, which contains human pathogenic species [16], also increased from wheat amendment. This genus has the capacity to digest starch and may have increased after bacteria utilized starch in the wheat for growth.

DON-degrading capacity has previously been shown to vary by soil type [17]. Despite the Lethbridge soil samples coming from plots that previously grew barley infected with *F*. *graminearum*, only soil samples from Lacombe exhibited catabolic activity against DON. The LA bacterial consortium had a maximum degradation of 26.4% after two passages in M9 medium. After the third passage, the consortium lost the ability to degrade DON. This may have been due to die-off of strains that could metabolize DON. Alternatively, although DON was the only carbohydrate in the medium, nutrients from the soil may have sustained growth in the initial passages, but their depletion limited bacterial growth in the third passage. If this was the case, DON may have been metabolized through an intrinsic mechanism and was not used as a carbohydrate nutrient by the consortium.

When the LA consortium was compared to LA potted soil, several genera were strongly increased in abundance, including *Pseudomonas*, *Delftia*, *Acinetobacter*, and *Stenotrophomonas*. None of these taxa were part of the ten most abundant genera in LA soil on day 0, although both *Pseudomonas* and *Stenotrohpomonas* increased in abundance in wheat-amended soil on day 7. It is notable that these same genera have been reported in bacteria consortia capable of degrading DON previously [3,4,5,9,18,19,20]. This suggests that members of these genera have a role in DON degradation in soil environments.

In order to isolate the strains of bacteria that could degrade DON from the LA consortium, bacteria from the consortium were identified and cultured with DON. *Pseudomonas* had the largest increase in abundance when the consortium was cultured with DON, and the majority of isolates on culture plates were *P*. *aeruginosa*; thus, most isolates evaluated for DON degradation were *P. aeruginosa*. We also tested *S*. *liquefaciens* because members of this genus only increased in wheat-amended LA soil, which had DON-degrading capacity, and not LE soil types. However, isolates of these species did not degrade DON. It is possible that *Pseudomonas* or *Serratia* from the LA consortia may have been involved in DON degradation through cooperative catabolism with other strains. Accordingly, it can be difficult to isolate a DON-degrading strain from soil due to complex microbial interactions that can affect growth and activity [19]. For example, the dual-member activity of *Pseudomonas* and *Devosia* spp. from *Tenebrio molitor* larval feces have been reported to transform DON [21]. While the successful isolation of single microorganisms from the soil with DON-transformation capabilities have been reported [4,22,23], the biodegradation of DON by soil organisms is likely strain-specific [24,25], making it challenging to isolate the individual bacteria that utilize DON as a carbon source or degrade it in a complete media; thus, future work isolating DON-degrading bacteria would benefit from testing highly diverse soil types.

## 4. Conclusions

This study revealed a shift in the bacterial microbiota in soil amended with DON-contaminated wheat, although changes were more likely due to the degradation of the wheat rather than DON. Out of three diverse soil types evaluated, only one (LA) displayed capacity to degrade DON. When a consortium of LA soil was further evaluated, several genera, including *Pseudomonas*, *Delftia*, *Acinetobacter*, and *Stenotrophomonas*, strongly increased when DON was the only carbohydrate source, and it is notable that these bacteria have been implicated in DON degradation previously. However, DON catabolism was incomplete and serial passages of the consortium lost their ability to degrade DON, suggesting that its use as a carbon source was limited. Evaluation of *P*. *aeruginosa* and *S*. *liquefaciens* strains from the consortium failed to identify the individual strains capable of degrading DON. Future work should include highly diverse soil types for screening communities for DON degradation to increase the chance of identifying the bacteria capable of metabolizing this mycotoxin.

## 5. Materials and Methods

### 5.1. Soil Samples, Culture Media, and Reagents

Soil was collected from three separate plots in September, 2023. On all three soil plots, barley (*Hordeum vulgare* L.) was grown and harvested prior to soil collection. Lacombe soil was classified as black chernozem and Lethbridge soil was classified as dark brown chernozem. Lethbridge soils were collected from plots that received manure fertilizer yearly (LE-MA) or non-manured plots (LE-NM), as described by [10]. Barley grown in Lethbridge was verified to be contaminated with *F*. *graminearum* during that harvesting year, while Lacombe barley was not contaminated. For each soil plot, 12 core samples (core size 240–660 cm^3^ for the 0–15 cm depth) were randomly collected and thoroughly mixed to form 1 composite sample. These composite samples were used as inoculant for the in vitro incubations. Prior to the study, soil pH and moisture content were analyzed, as described by Ellert and Rock [26], using the Automated elemental microanalyzer (Thermo Flash 2000 EA, Milan, Italy).

Minimal medium (M9 broth; Sigma-Aldrich, Oakvile, ON, Canada) and Bacto tryptic Soy broth (BTSB; Becton Dickinson and Company BD Diagnostic Systems, Sparks MD, US) were used for consortia growth and bacterial isolation. M9 broth was supplemented with 0.02% (*w*/*v*) MgSO_4_, 0.001% (*w*/*v*) CaCl_2_ and 10 µg/mL of DON as a sole carbon source. Solid M9 was supplemented with 1% agar. Purified DON (≥98%) was obtained from Toronto Research Chemicals, Brisbane Road Toronto, ON, Canada, and used for supplementing the growth media. Concentration of DON was measured using a commercially available ELISA kit (Romer Labs Division Holding GmBH, AgraQuant, 3430 Tulin, Austria). The soil from LA, LE-MA and LE-NM was used for microbial inoculant in the in vitro cultures.

### 5.2. In Vitro Soil Cultures and Isolation of DON-Degrading Consortia

Triplicate soil samples (180 g) were placed directly into 10 cm pots (control) or amended with 20 g of unground DON-contaminated wheat (18 ppm/kg; 10% *w*/*w*) prior to pot placement. Briefly, 25 g of unground DON-contaminated wheat was weighed into a 500 mL Erlenmeyer flask and dissolved with 200 mL of deionized water on shaker for 30 min. The extract was filtered using a Whatman membrane paper (0.4 µm). The extract was processed using the DonStar^TM^ R—Immunoaffinity column (Romer Labs Diagnostic GmbH, Technopark 1, 3430 Tulin, Austria) and based on the manufactural protocol. Eluded DON was quantified using the DON plus ELSA kit. The wheat-soil mixture was homogenized, and the pots were incubated in a controlled environment (22 °C, 14 h light per day) for 32 d to detect short- and long-term microbial activity. Each pot was irrigated using adjustable-flow drip stakes connected to a central drip irrigation system. Water was delivered at regular intervals based on the observation of soil moisture, maintaining an optimal volumetric water content of 30–50% in the soil mixture. On days 0, 7, 14, and 32, subsamples (approximately one gram) were collected from each pot, and serially diluted 10-fold in a M9 or BTSB containing DON (10 µg/mL). The dilutions were incubated for 7 d at 30 °C under aerobic conditions with shaking (300 rpm). After 7 d of incubation, 200 µL of dilution was transferred into 1.8 mL of M9 or BTSB supplemented with 10 µg/mL DON for a second serial passage, followed by incubation for another 7 d under same the conditions. Bacterial isolates obtained from the second serial passage incubation were cultivated on solid media and tested for DON degradation in M9 broth containing 10 µg/mL of DON as sole carbon source as well as in BTSB. Isolates were incubated for 24 h in aerobic conditions with shaking (300 rpm). Degradation of DON in the supernatant of the second serial passage incubation (*N* = 3) and in cultures of the pure isolates were evaluated using the previously described commercial ELISA kit. Soil from the pots at each time point, and consortia that degraded DON, were stored at −80 °C for DNA extraction.

### 5.3. DNA Isolation and 16S rRNA Sequencing

Metagenomic DNA was extracted from soil samples and consortia using a DNeasy Powerlyzer PowerSioil Kit (QIAGEN, Hilden, Germany), according to the manufacturer’s instructions. For the consortia, bacteria were pelleted by centrifugation. The DNA concentration was measured and then normalized to 10 ng/µL. The normalized DNA was used for PCR with barcode primers for 25 cycles. All samples after PCR were normalized into the same fmol, pooled together, and ligated for 16S rRNA full-length genome sequencing using a nanopore MinION (PromethION 2 Solo) with flonge flowcell (R10.4). The base calling used a super-high-accuracy method with an NVIDIA GPU (RTX3090). The OTU and taxonomy table was generated with nanopore wf-16S pipeline with base-called raw reads. Similarly, DNA was extracted from the microbial consortia as well as from the single isolates using the DNeasy Blood and Tissue Kit (QIAGEN, ON, Canada). The 16S rRNA gene was amplified by PCR with primers 27F (5′-AGAGTTTGATCMTGGCTCAG-3′) and 1492R (5′-CGGTTACCTTGTTACGACTT-3′) for both single bacteria and consortia, and was sequenced by nanopore MinION with a flonge flowcell (R10.4).

### 5.4. Analysis of Bacterial Community Diversity from Different Environmental Sources and Consortia

Raw reads were assessed using FastQC 0.11.9, and a summary report was produced using MultiQC 1.12 [27]. Trimmomatic (v0.39) was used to perform sequence trimming [28] to eliminate primer sequences and low-quality regions. Subsequent statistical analysis was performed using R 4.1.0 [29]. The trimmed reads underwent additional filtering using DADA2 1.22.0 [30] filterAndTrim with default settings before being merged. Bimeric sequences were removed from the merged dataset utilizing DADA2′s. Taxonomic classification of the 16S sequences was conducted using DADA2 IdTaxa algorithm with the SILVA 138 database [31] to generate an amplicon sequence variants (ASVs) table. Statistical analysis and visualization of the data were conducted using Phyloseq 1.38.0 [32], vegan 2.5–6 [33], DESeq2 1.34.0 [34], and ggplot2 3.3.5 [35]. Alpha diversity, measured using the Shannon index and observed ASVs richness, was analyzed and visualized using vegan and plotted. The ASV table underwent filtering to include only ASVs present (count ≥ 2) in at least 1% of the samples to minimize downstream noise. The effects of treatment and time on alpha diversity were evaluated using a one-way ANOVA. To evaluate the treatment effect, microbial community structure was examined using β-dispersion and PERMANOVA (9999 permutations) implemented in vegan 2.5–6 [33]. Beta diversity was assessed by normalizing filtered ASV counts using GMPR-derived size factors within the DESeq2. Bray–Curtis metric was employed to determine sample-sample distances, followed by visualization using detrended correspondence analysis (DCA). The most abundant phyla and genera for each treatment and sample time were identified using phyloseq and visualized. Differentially abundant genera (*p* < 0.01) or (*p* < 1×10^−4^) were identified utilizing DESeq2 by fitting a negative binomial model to each comparison of variables.

## Figures and Tables

**Figure 1 toxins-17-00565-f001:**
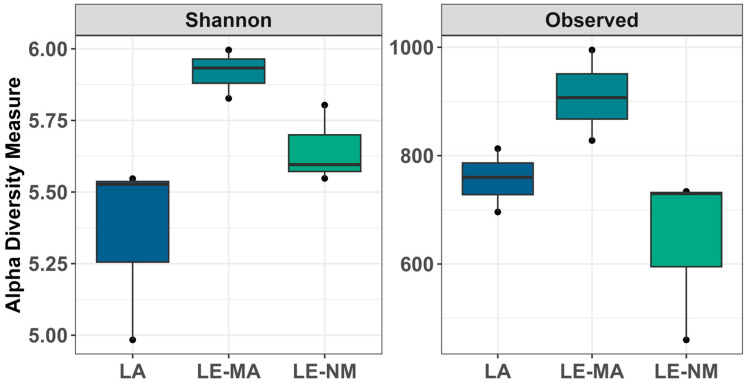
Alpha diversity showing Shannon Diversity and bacterial richness (observed taxa) in soil samples at day 0. The box in the plots indicates the interquartile range (IQR; middle 50% of the data), the middle line represents the median value, and the whiskers represents 1.5 times the IQR. Lacombe (LA) samples originated from central Alberta. LE samples were from Lethbridge in southern Alberta and included plots fertilized with manure (MA) or not fertilized (NM).

**Figure 2 toxins-17-00565-f002:**
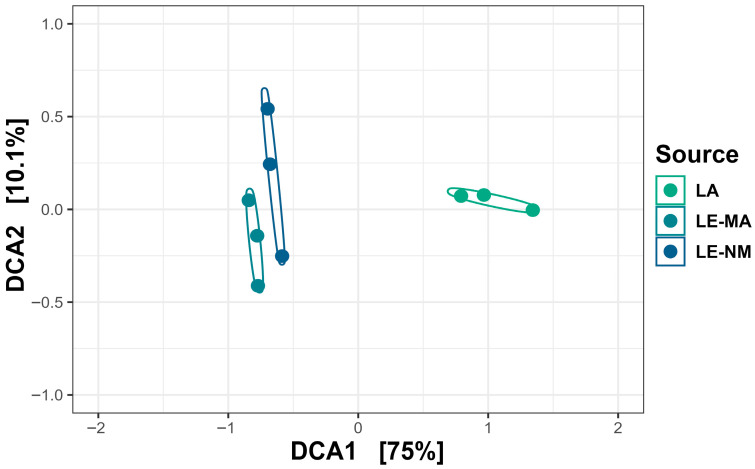
Detrended correspondence analysis (DCA) of the Bray–Curtis metric showing clustering by soil source at day 0. LA (Lacombe) samples originated from central Alberta. LE (Lethbridge) samples were from southern Alberta and included plots fertilized with manure (MA) or not fertilized (NM).

**Figure 3 toxins-17-00565-f003:**
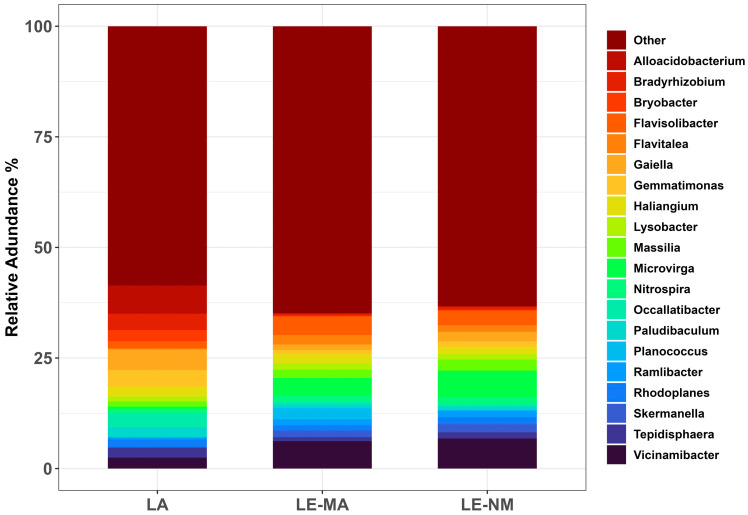
Relative abundance of the ten most-abundant genera from each soil source on day 0. Other, accounts for genera outside the top ten for each soil type. Lacombe (LA) samples originated from central Alberta. Lethbridge (LE) samples were from southern Alberta and included plots fertilized with manure (MA) or not fertilized (NM).

**Figure 4 toxins-17-00565-f004:**
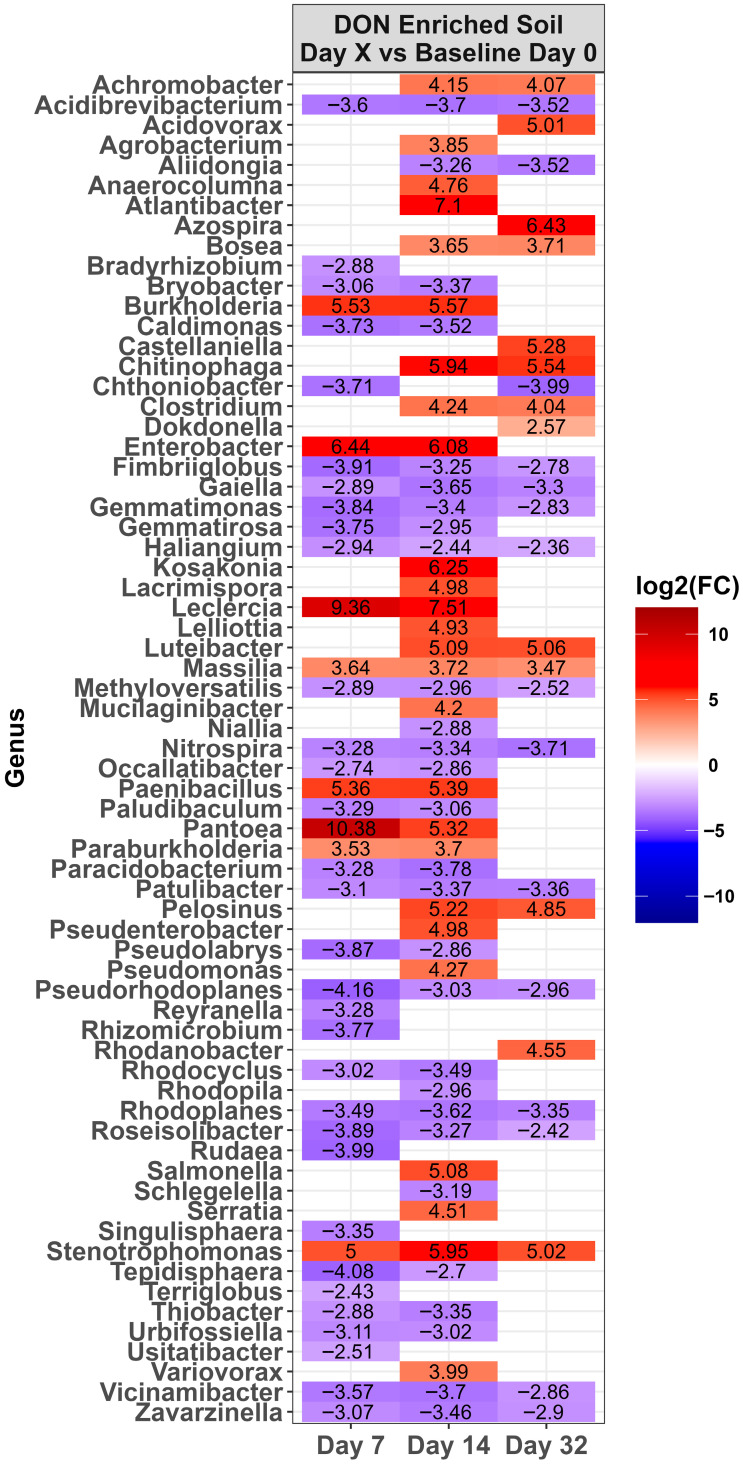
Heat map of the relative abundance of the genera in the Lacombe (LA) soil that was amended with DON-contaminated wheat. Genera that showed a significant change [*p* < 0.0001] compared to the baseline (day 0) across different time points are shown. The colors displayed represent the average log2(FC) of amplicon sequence variants (ASVs) with a significant change (*p* < 0.0001) within the respective genera at the indicated time. Note: The LA control soil did not change in genera across time, and is therefore not shown. This is example 2 of an equation.

**Figure 5 toxins-17-00565-f005:**
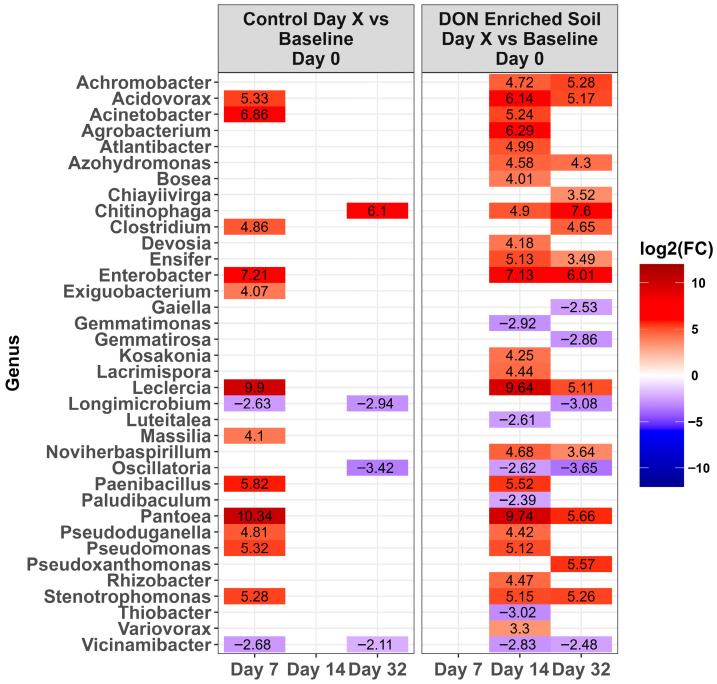
Heat map of the relative abundance of the genera in the Lethbridge manured (LE-MA) soil that was amended with DON-contaminated wheat (DON-enriched) or not amended (control). Genera that showed a significant change [*p* < 0.0001] compared to the baseline (day 0) across different time points are shown. The colors displayed represent the average log2(FC) of amplicon sequence variants (ASVs) with a significant change (*p* < 0.0001) within the respective genera at the indicated time.

**Figure 6 toxins-17-00565-f006:**
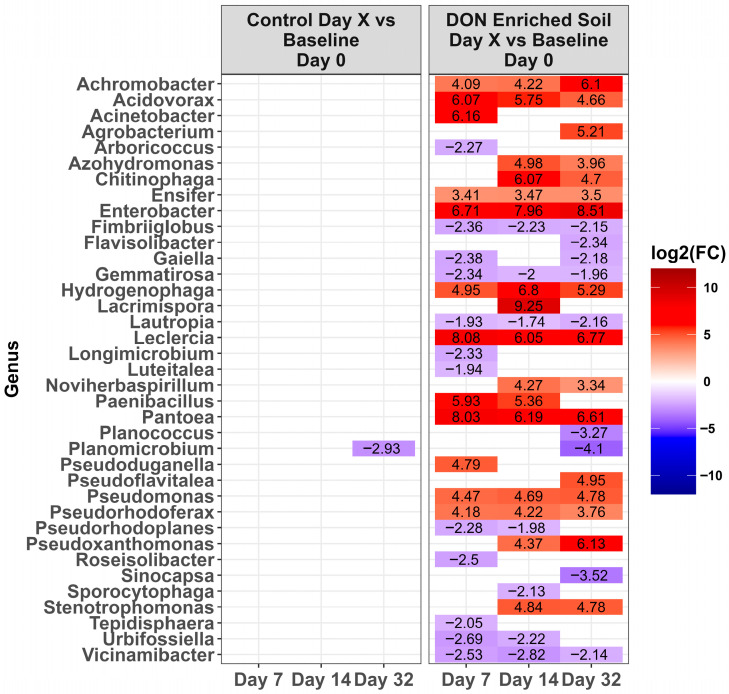
Heat map of the relative abundance of the genera in the Lethbridge non-manured (LE-NM) soil that was amended with DON-contaminated wheat (DON-enriched) or not amended (control). Genera that showed a significant change [*p* < 0.0001] compared to the baseline (day 0) across different time points are shown. The colors displayed represent the average log2(FC) of amplicon sequence variants (ASVs) with a significant change (*p* < 0.0001) within the respective genera at the indicated time.

**Figure 7 toxins-17-00565-f007:**
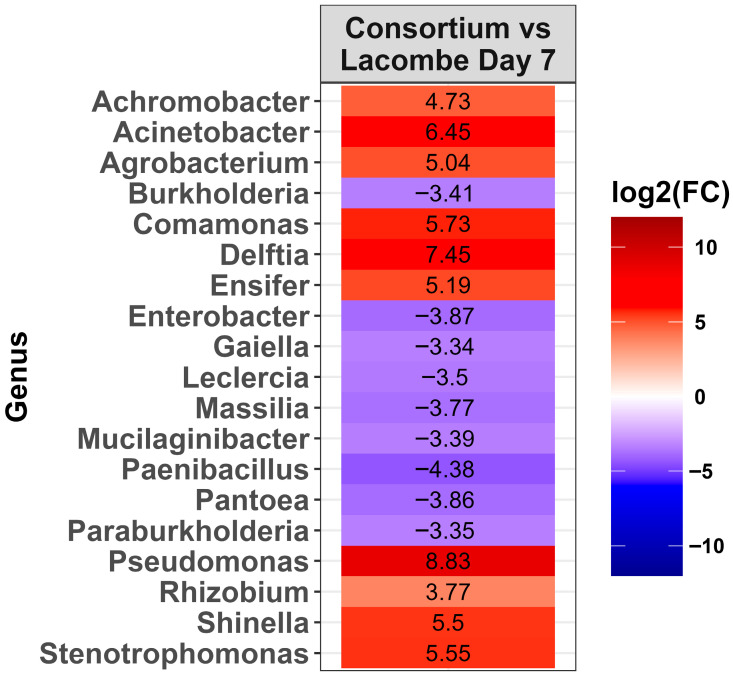
Heat map of the relative abundance of the genera in the Lacombe (LA) consortium that showed a significant change [*p* < 0.0001] compared to Lacombe day 7 potted soil. The colors displayed represent the average log2(FC) of amplicon sequence variants (ASVs) with a significant change (*p* < 0.0001) within the respective genera.

## Data Availability

The original contributions presented in this study are included in the article/Supplementary Material. Further inquiries can be directed to the corresponding author(s).

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
