# Peer review of "Bacterial Microbiota in Soil Amended with Deoxynivalenol-Contaminated Wheat"

_toxins, 2025, doi:10.3390/toxins17120565_

Round 1

Reviewer 1 Report

Comments and Suggestions for Authors

In the paper entitled: "The Bacterial Microbiota in Soil Amended with Deoxynivalenol-Contaminated Wheat," the authors analyze soil samples to evaluate the capacity of the microorganisms present to degrade the mycotoxin deoxynivalenol.

This work confirms that in soils dedicated to agricultural cultivation, it is possible to find groups or consortia of microorganisms, which are part of the microbiota, that naturally have the capacity to degrade toxic compounds such as mycotoxins produced by different Fusarium species.

The presence of mycotoxins in grains of crops such as wheat or barley is well documented and constitutes a significant public health problem due to the consequences that the consumption of these mycotoxins has on human health. Therefore, all studies aimed at elucidating the presence of microorganisms are of great importance.

This paper is well-written, both the abstract and the introduction. In the latter, the objectives of the work are clear and consistent with the experimental design described in the Materials and Methods section.

However, when the authors refer to DON degradation, the results would be more conclusive if, in addition to detecting DON with the ELISA kit, they had confirmed which degradation products were forming in the soil samples collected from Lacombe, Louisiana. The authors mention in the introduction that the degradation products produced by microorganisms can be de-epoxy-DON (DOM-1) or 3-keto-DON, the latter exhibiting moderate toxicity levels.

The question that arises is, why didn't the authors analyze the compounds produced by DON degradation by the microorganisms present in the samples from Lacombe, Louisiana, using a mass spectrometer?

Author Response

Response to Reviewer One

Reviewer ONE (Comments and Suggestions for Authors)

In the paper entitled: "The Bacterial Microbiota in Soil Amended with Deoxynivalenol-Contaminated Wheat," the authors analyze soil samples to evaluate the capacity of the microorganisms present to degrade the mycotoxin deoxynivalenol.

This work confirms that in soils dedicated to agricultural cultivation, it is possible to find groups or consortia of microorganisms, which are part of the microbiota, that naturally have the capacity to degrade toxic compounds such as mycotoxins produced by different Fusarium species.

The presence of mycotoxins in grains of crops such as wheat or barley is well documented and constitutes a significant public health problem due to the consequences that the consumption of these mycotoxins has on human health. Therefore, all studies aimed at elucidating the presence of microorganisms are of great importance.

This paper is well-written, both the abstract and the introduction. In the latter, the objectives of the work are clear and consistent with the experimental design described in the Materials and Methods section.

However, when the authors refer to DON degradation, the results would be more conclusive if, in addition to detecting DON with the ELISA kit, they had confirmed which degradation products were forming in the soil samples collected from Lacombe, Louisiana. The authors mention in the introduction that the degradation products produced by microorganisms can be de-epoxy-DON (DOM-1) or 3-keto-DON, the latter exhibiting moderate toxicity levels.

The question that arises is, why didn't the authors analyze the compounds produced by DON degradation by the microorganisms present in the samples from Lacombe, Louisiana, using a mass spectrometer?

RESPONSE: Thank you for the question. Mass spectrometer would be relevant to determine the degradation products produced in this study. However, a maximum degradation of 26.4% was observed. The concentration of the degradation products may be below the detection level and difficult to distinguish from other metabolites. Therefore, we focused on the bacterial microbiota composition in the soils amended with DON-contaminated wheat. We intend to incorporate mass spectrometry in our future studies with higher degradation efficiency.

Reviewer 2 Report

Comments and Suggestions for Authors

The manuscript "The Bacterial Microbiota in Soil Amended with Deoxynivalenol-contaminated Wheat" presents a valuable investigation into the response of diverse soil microbiomes to DON contamination. The study analyzed the community structure and composition of soil bacteria, changes in the microbiota across time, and DON degrading ability of three diverse soil types, one sample from central (Lacombe, Alberta) and two samples (a manure-amended sample and a non-manure-amended sample) from southern (Lethbridge, Alberta). Although the study failed to identify individual strain capable of degrading DON, it lays a solid foundation for future work. Several methodological and presentation issues need to be addressed to strengthen the manuscript's clarity, reproducibility, and impact.

Major Concerns:

  1. Soil Sampling Rationale and Design:
    • The choice of only three soil samples—specifically, one from Lacombe with no manure history and two (manured and non-manured) from Lethbridge—is a key aspect of the study design. The Introductionshould explicitly state the rationale for this selection. Please clarify the hypothesis behind comparing a single sample from one location to a paired set from another. This will help readers understand whether the primary comparison is between geographic locations or manure amendment histories.
  2. DON Amendment Methodology:
    • (Line 294)The use of "20 g of unground DON-contaminated wheat (18 ppm/kg)" raises a significant concern regarding homogeneity and reproducibility. Using unground material makes it difficult to ensure a uniform distribution of the toxin (DON) across all experimental pots. Please describe how the 18 ppm/kg concentration was verified for the specific batch of unground wheat used and explain the steps taken to ensure consistent DON application to each replicate. If the wheat was homogenized prior to portioning, this should be stated explicitly.

Minor Concerns:

Introduction:

  • As mentioned above, please provide a clearer justification for the soil sampling strategy to frame the study's objectives more effectively.

Materials and Methods:

  • (Line 286)Please specify the purity and supplier of the purified DON used in the liquid culture assays.
  • (Line 299)The sampling time points (days 0, 7, 14, and 32) should be justified. The gap between day 14 and day 32 is notably longer than the previous weekly intervals. Please provide a brief rationale for selecting day 32 instead of a more consistent interval (e.g., day 28) to demonstrate a logical sampling scheme that captures both short-term and long-term dynamics.
  • (Line 300)For clarity and reproducibility, please specify the dilution factor used in the serial dilution procedure (e.g., "serially diluted 10-fold in...").

Results:

  • (Line 172)Please only italicize the genus and species names of bacteria in the sentence.

Author Response

Response to Reviewer Two

Reviewer TWO (Comments and Suggestions for Authors)

The manuscript "The Bacterial Microbiota in Soil Amended with Deoxynivalenol-contaminated Wheat" presents a valuable investigation into the response of diverse soil microbiomes to DON contamination. The study analyzed the community structure and composition of soil bacteria, changes in the microbiota across time, and DON degrading ability of three diverse soil types, one sample from central (Lacombe, Alberta) and two samples (a manure-amended sample and a non-manure-amended sample) from southern (Lethbridge, Alberta). Although the study failed to identify individual strain capable of degrading DON, it lays a solid foundation for future work. Several methodological and presentation issues need to be addressed to strengthen the manuscript's clarity, reproducibility, and impact.

Major Concerns:

  1. Soil Sampling Rationale and Design:
    • Comment: The choice of only three soil samples—specifically, one from Lacombe with no manure history and two (manured and non-manured) from Lethbridge—is a key aspect of the study design. The Introduction should explicitly state the rationale for this selection. Please clarify the hypothesis behind comparing a single sample from one location to a paired set from another. This will help readers understand whether the primary comparison is between geographic locations or manure amendment histories.
    • Response: We appreciate the reviewer’s comment regarding manure and non-manure amended soil samples used in the study. We stated in line 53-54 that “Microbial communities from various environmental sources may have a distinct response to DON enrichment”. Our rational for soil samples from different location Lacombe and Lethbridge as well as manure-amended was to increase the chances of obtaining a diverse microbial community with ability to degrade DON.
  2. DON Amendment Methodology:
    • Comment: (Line 294) The use of "20 g of unground DON-contaminated wheat (18 ppm/kg)" raises a significant concern regarding homogeneity and reproducibility. Using unground material makes it difficult to ensure a uniform distribution of the toxin (DON) across all experimental pots. Please describe how the 18 ppm/kg concentration was verified for the specific batch of unground wheat used and explain the steps taken to ensure consistent DON application to each replicate. If the wheat was homogenized prior to portioning, this should be stated explicitly.
    • Response: We appreciate the reviewer’s comment. We have included the description on how we extracted and quantified DON in the DON-contaminated wheat in line 296-300. “A 25 g of unground DON-contaminated wheat was weighed into a 500 mL Erlenmeyer flask and dissolved with 200 mL of deionized water on shaker for 30 min. The extract was filtered using a whatman membrane paper (0.4 µm). The extract was processed using the DonStarTM R - Immunoaffinity column and based on the manufactural protocol. Eluded DON was quantified using the DON plus ELSA kit”.
    • Response: The sentence has been rephrased in line 301 to indicate that the wheat-soil mixture was homogenized prior to the pots incubated.

Minor Concerns:

Introduction:

  • Comment: As mentioned above, please provide a clearer justification for the soil sampling strategy to frame the study's objectives more effectively.
  • Response: Thank you for the reviewer’s comment. As stated above, we hypothesized that diverse soil samples from different location and manure-amended could harbor microbial community with DON degrading ability line 59-60. “The present study aimed to identify microbial consortia in diverse soil samples, from central (Lacombe, LA) and southern (Lethbridge, LE) Alberta that could degrade DON”

Materials and Methods:

  • Comment: (Line 286) Please specify the purity and supplier of the purified DON used in the liquid culture assays.
  • Response: The purity DON (≥98%) has been added in line 286.
  • Comment: (Line 299) The sampling time points (days 0, 7, 14, and 32) should be justified. The gap between day 14 and day 32 is notably longer than the previous weekly intervals. Please provide a brief rationale for selecting day 32 instead of a more consistent interval (e.g., day 28) to demonstrate a logical sampling scheme that captures both short-term and long-term dynamics.
  • Response: We appreciate the reviewers comment regarding capturing both short- and long-term dynamics. That is what we did and day 32 was to capture durable changes beyond the normal two weeks interval. The sentence in line 295-296 now states that “The pots were incubated in a controlled environment (22°C, 14 h light per day) for 32 d to detect short- and long-term microbial activity”.
  • Comment: (Line 300) For clarity and reproducibility, please specify the dilution factor used in the serial dilution procedure (e.g., "serially diluted 10-fold in...").
  • Response: Thank you, the sentence has been corrected as suggested.

Results:

  • Comment: (Line 172) Please only italicize the genus and species names of bacteria in the sentence.
  • Response: The correction has been affected in this line.

Reviewer 3 Report

Comments and Suggestions for Authors

The submitted paper presents interesting results. After overall assessment and minor revisions, I recommend it for publication as a short communication.

On days 0, 7, 14 and 32, subsamples (approximately one gram) were  collected from each pot, and serially diluted in a M9 or BTSB containing DON (10 μg/mL).

  • How was sample representativeness ensured given that only 1 g was taken?
  • Were the results affected by the sampling procedure?

The authors note that shifts in the bacterial microbiota were likely driven by wheat degradation rather than DON. In line with this, the manuscript lacks a wheat-only variant/control. Including such a control (or explicitly discussing its absence and implications) would strengthen the study, as its omission is a notable weakness of the presented results.

There are also formal issues with scientific nomenclature. Latin names for microorganisms and plants are not consistently italicized throughout (i.e., not always).

Author Response

Response to Reviewer Three

Reviewer THREE (Comments and Suggestions for Authors)

The submitted paper presents interesting results. After overall assessment and minor revisions, I recommend it for publication as a short communication.

Response: We appreciate the suggestion. The volume of data generated in this study is beyond the scope of a short communication. This work describes microbiota diversity in the soils amended with DON-contaminated wheat and our findings, which we were able to present in a full text format, provide a solid foundation for future studies. We would like to pursue a full research article.

Comment: On days 0, 7, 14 and 32, subsamples (approximately one gram) were collected from each pot, and serially diluted in a M9 or BTSB containing DON (10 μg/mL). How was sample representativeness ensured given that only 1 g was taken?

Response: Thank you for the question. 1 g was taken from triplicate samples according to standard procedure to ensure representativeness.

Comment: Were the results affected by the sampling procedure?

Response: We appreciate the question that sampling procedure can affect results. We used standard procedures whereby 1 g of soil was taken from triplicate samples to ensure representativeness of the microbial diversity.

Comment: The authors note that shifts in the bacterial microbiota were likely driven by wheat degradation rather than DON. In line with this, the manuscript lacks a wheat-only variant/control. Including such a control (or explicitly discussing its absence and implications) would strengthen the study, as its omission is a notable weakness of the presented results.

Response: We appreciate the reviewer’s comment. We mentioned the lack of a non-DON-contaminated wheat in this study in line 210-211 as a limitation in the study. We focused on the microbial community of soils amended with DON-contaminated wheat by comparing with uncontaminated subsamples from the soil. We could adjust this in our future study to compare the differences between uncontaminated wheat to strengthen our findings.

Comments: There are also formal issues with scientific nomenclature. Latin names for microorganisms and plants are not consistently italicized throughout (i.e., not always).

Response: All scientific names have been checked and italicized throughout the text as recommended.

Round 2

Reviewer 2 Report

Comments and Suggestions for Authors

The authors have mostly addressed my concerns. I am satisfied with the manuscript and am happy to accept it in its current form.